# The Relationship between Oxidative Stress and Subjective Sleep Quality in People with Coronary Artery Disease

**DOI:** 10.3390/brainsci12081070

**Published:** 2022-08-12

**Authors:** Vivian Feng, Shankar Tumati, Ruoding Wang, Kritleen K. Bawa, Damien Gallagher, Nathan Herrmann, Susan Marzolini, Paul Oh, Ana Andreazza, Krista L. Lanctôt

**Affiliations:** 1Neuropsychopharmacology Research Group, Sunnybrook Research Institute, 2075 Bayview Avenue, Room FG21, Toronto, ON M4N 3M5, Canada; 2Department of Pharmacology and Toxicology, University of Toronto, Toronto, 1 King’s College Cir, Toronto, ON M5S 1A8, Canada; 3Department of Psychiatry, Temerty Faculty of Medicine, University of Toronto, 1 King’s College Cir, Toronto, ON M5S 1A8, Canada; 4Geriatric Psychiatry, Sunnybrook Health Sciences Centre, Toronto, ON M4N 3M5, Canada; 5Evaluative Clinical Sciences, Hurvitz Brain Sciences Program, Sunnybrook Research Institute, Toronto, ON M4N 3M5, Canada; 6KITE Toronto Rehabilitation Institute, University Health Network, East York, ON M5G 2A2, Canada; 7Centre for Addictions and Mental Health, 1000 Queen Street West, Toronto, ON M6J 1H4, Canada

**Keywords:** sleep quality, oxidative stress, coronary artery disease

## Abstract

Background: (1) Sleep disorders are prevalent in coronary artery disease (CAD) patients and predict cardiac events and prognosis. While increased oxidative stress (OS) has been associated with sleep disorders, less is known about its relationship with sleep quality. Similarly, little is known of how this relationship might change with exercise, which can improve sleep quality. Factors of sleep quality, such as sleep duration and disturbances, are also important as they predict cardiovascular diseases better than a global score alone. This study investigated whether OS was associated with self-rated sleep quality and its factors before and after completing a 24-week exercise intervention. (2) Methods: CAD patients undergoing an exercise program were recruited. OS was measured at baseline by the concentrations of early- (lipid hydroperoxides, LPH) and late-stage (8-isoprostane, 8-ISO) lipid peroxidation products and their ratio. Sleep quality was measured by the self-reported Pittsburgh Sleep Quality Index (PSQI) instrument at baseline and termination. Three sleep factors—perceived sleep quality, sleep efficiency, and daily disturbances—were derived from the PSQI. (3) Results: Among CAD patients (*n* = 113, 85.0% male, age = 63.7 ± 6.4 years, global PSQI = 5.8 ± 4.0), those with poor sleep (PSQI ≥ 5) had higher baseline 8-ISO levels (F(1, 111) = 6.212, *p* = 0.014, η_p_^2^ = 0.053) compared to those with normal sleep. Concentrations of LPH (F(1, 105) = 0.569, *p* = 0.453, η_p_^2^ = 0.005) and 8-ISO/LPH ratios (F(1, 105) = 2.173, *p* = 0.143, η_p_^2^ = 0.020) did not differ between those with poor sleep and normal sleep. Among factors, perceived sleep quality was associated with 8-ISO and 8-ISO/LPH, and daily disturbances were associated with 8-ISO. (4) Conclusions: A marker of late-stage lipid peroxidation is elevated in CAD patients with poor sleep and associated with daily disturbances, but not with other factors or with sleep quality and its factors after exercise intervention.

## 1. Introduction

Sleep-wake disturbances are strongly associated with cardiovascular diseases, such as coronary artery disease (CAD), and cardiovascular risk factors, including hypertension, diabetes, obesity, and smoking [1,2]. The mechanisms underlying the relationship between sleep and cardiovascular disease are not known; oxidative stress (OS), which is increased in CAD and in healthy adults with poor sleep, may be involved [3,4,5,6].

OS results from an imbalance between reactive oxygen species (ROS) production and antioxidants, and induces lipid, deoxyribonucleic acid, and protein damage in the body. Lipid membranes are particularly susceptible to oxidation by free radicals [7,8,9]. Termed lipid peroxidation, this process produces lipid hydroperoxides (LPH), which readily decompose [7,9] and are eliminated by antioxidants under normal circumstances. Prolonged excess LPH results in late-stage products such as isoprostanes, which are stable and the most reactive when inducing free radical damage [10]. The late by-product 8-isoprostane (8-ISO) circulates in the plasma, is excreted in urine [11], and is robust against renal and hepatic activity, allowing for the precise quantification of OS in vivo [12]. As lipid peroxidation cannot be repaired, even moderate levels of lipid peroxidation have physiological consequences for cell signaling and membrane remodeling [10].

OS is linked to the pathophysiological mechanisms underlying CAD [13,14,15,16,17,18]. Higher levels of lipid peroxidation markers, specifically 8-ISO, are found in those with cardiovascular conditions [19,20] and in healthy individuals with cardiovascular risk factors, such as smoking, hypercholesterolemia, obesity, and diabetes [21]. OS is also linked to negative outcomes in CAD [14] presumably by impairing endothelial function and activating localized inflammation [22].

Most clinical studies analyzing the relationship between sleep and CAD used a global sleep score, which may mask specific sleep factors that contribute to overall sleep quality [23,24]. Assessing multiple factors of sleep quality, such as sleep duration and disturbances, may be important as they have been found predict cardiovascular diseases better than a global score [25,26].

It has been established that exercise programs improve cardiorespiratory fitness, have moderate benefits for sleep symptoms in middle-aged adults [27], and reduce the risk of cardiovascular events in those with cardiovascular risk factors, such as obesity [28]. OS may be a mechanistic link between poor sleep and exercise as both are correlated with CAD [29,30,31].

It is currently not known whether sleep quality in CAD is associated with OS, particularly lipid peroxidation. Understanding this relationship may enable a better clinical characterization of patients and identify therapeutic targets. This study aims to quantify the relationship between peripheral biomarkers of lipid peroxidation in CAD patients and self-reported sleep quality. We hypothesized that concentrations of lipid peroxidation biomarkers will be elevated in poor sleepers compared to normal sleepers. We further investigated if biomarkers of OS measured at baseline were associated with sleep factors (Perceived Sleep Quality, Sleep Efficiency, and Daily Disturbances) before and after an exercise program.

## 2. Materials and Methods

### 2.1. Participants

Eligible participants were recruited from a cardiac rehabilitation program and provided written informed consent. All study participants had a history of CAD (defined as previous history of either myocardial infarction, coronary artery bypass graft, percutaneous transluminal coronary angioplasty or at least 50% stenosis in 1 or more major coronary arteries). All participants were aged between 45 and 85 years and spoke and understood English. Participants were excluded if they were previously diagnosed with neurodegenerative illnesses, a neurological disorder, unstable angina, or substance abuse, or were women of childbearing age. Individuals with ventricular tachycardia and/or an implantable cardioverter defibrillator were also excluded. The presence of a major or minor depressive episode was assessed at baseline using standardized criteria from the Structured Clinical Interview for Diagnostic and Statistical Manual of Mental Disorders, Fourth Edition. Antidepressant use was permitted if used at a stable dose for at least 3 months before the trial.

### 2.2. Study Design

Data were obtained from a prospective study that investigated biomarkers that were associated with change in cognitive performance in CAD patients (ClinicalTrials.gov NCT01625754).

### 2.3. Cardiac Rehabilitation Program

Exercise prescriptions were based on cardiopulmonary assessments developed by the cardiac rehabilitation program, and consisted of supervised aerobic and resistance training each week for 24 weeks [32]. A typical supervised class began with a half hour lecture, followed by 15 min of warm-up and stretching exercises, and individualized exercise prescriptions. The exercise training was of moderate intensity and lasted approximately 2 h. Initial prescriptions had an intensity equivalent to 60% of peak oxygen uptake (VO2max; approximate distance of 1.6 km) and prescriptions were advanced biweekly, increasing the distance to a maximum of 6.4 km and increasing intensity to a maximum of 80% VO2max (maximum duration of 60 min), where exercise training was maintained thereafter. High-risk patients, including those with chronic heart failure, cardiac transplantation, complex ventricular ectopy, symptomatic ST segment depression, or atrial fibrillation, were scheduled initially for up to 5 weekly supervised sessions, which were reduced to one or two sessions when warranted [32].

### 2.4. Clinical Characteristics

Participant interviews and chart reviews were used to collect demographic information and medical history, including cardiac diagnoses, concomitant cardiac and diabetes medications, and cardiovascular risk factors (i.e., hyperlipidemia, hypertension, etc.). Anthropometrics and fitness characteristics, including height, weight, body mass index (BMI), and waist circumference, were assessed by the exercise team at the cardiac rehabilitation site.

### 2.5. Measures

The Pittsburgh Sleep Quality Index (PSQI) global score was used to measure sleep quality [33]. The PSQI is a 19-item self-rated questionnaire that reliably assesses sleep status and disturbances over the past month in healthy and patient populations. The PSQI assesses seven aspects of sleep quality: subjective sleep quality, sleep latency, sleep duration, habitual sleep efficiency, sleep disturbance, use of sleep medication, and daytime dysfunction. A global sleep score (maximum score of 21) is obtained by summing each item score, conveying both the number and severity of sleep problems. Higher scores on the PSQI indicate worse sleep quality. Poor sleep quality was defined as scores greater than or equal to 5 as has been done previously [33]. The PSQI has internal consistency, reliability, and construct validity (convergent, discriminant, and known group validity) amongst various populations [34]. Factors were taken from a 3-factor model as previously described in older adults with depressive symptoms [35]. Sleep Efficiency (Factor 1) consisted of items 3 and 4; Perceived Sleep Quality (Factor 2) consisted of items 1, 2, and 6; and Daily Disturbances (Factor 3) consisted of items 5 and 7 [35]. The 3-factor scoring of the PSQI may reveal impairments in a single dimension that may be missed in a global score and provide clinically relevant information [35]. In addition, anxiety and depressive symptoms were assessed with the Hospital Anxiety and Depression Scale (HADS).

### 2.6. Samples

Biomarkers of OS were available at baseline. Fasting blood was obtained in the morning prior to exercise and centrifuged at 1000× *g* for 10 min at 4 °C and stored at −80 °C immediately after. OS was measured by quantifying the late-stage lipid peroxidation marker 8-ISO (Cayman Chemical; Item No. 516351) in blood by a standard competitive sandwich ELISA using an 8-ISO acetylcholinesterase conjugate as a tracer and 8-ISO-specific rabbit antiserum. The sensitivity of this assay is between 0.8 and 500 pg/mL. The intra-assay coefficient of variation is between 1% and 19%, and the inter-assay presented a coefficient of variation of 9.1%. Results were expressed as pg/mL and square-root-transformed to produce normality for analysis. LPH was assessed using the colorimetric LPH assay kit (Cayman Chemical; item 705002) according to the manufacturer’s instructions. The sensitivity of this kit is between 0.25 and 5 nmol hydroperoxides. The intra-assay coefficient of variation (CV) was between 2% and 15%, and the inter-assay coefficient of variation was 9%.

### 2.7. Statistical Analyses

All analyses were carried out in SPSS (version 27.0, IBM Inc. Armonk, NY, USA) and considered significant at a two-tailed α of <0.05. Continuous variables were summarized as means and standard deviations (SD). Categorical variables were expressed as numbers and percentages. Differences between participant characteristics with normal or poor sleep quality (as defined using PSQI cut-off of greater than or equal to 5) were assessed using an independent samples *t*-test for continuous variables and a chi-squared test for categorical variables. The primary hypothesis was analyzed using a univariate analysis of covariance (ANCOVA) to determine the difference in baseline 8-ISO between those with PSQI < 5 and PSQI ≥ 5. Separate multiple linear regression models were used to assess whether 8-ISO significantly predicted baseline and termination global PSQI scores. Exploratory analyses investigated the relationship between 8-ISO and 8-ISO/LPH and PSQI factor scores. Post hoc multiple linear regression models were used to investigate the relationship between 8-ISO and 8-ISO/LPH and termination PSQI factors. Adjusted models were exploratory and included demographic measures (age, sex, and BMI) and neuropsychiatric symptoms (anxiety and depression scores) as covariates. Longitudinal models were additionally adjusted for baseline PSQI score.

## 3. Results

### 3.1. Patient Characteristics

From an observational trial with 120 participants, data were obtained from 113 participants with baseline values for both serum OS concentrations and PSQI. Baseline demographics and clinical characteristics are shown in Table 1. The comparison of demographic factors between groups revealed that patients with poor sleep were more likely to be female, have a higher BMI, and have greater anxiety and depression scores (Table 1). Among those who completed the 24-week exercise intervention (*n* = 112), there were 96 participants with termination sleep scores. There was a significant improvement in sleep quality (global PSQI score; t(1, 94) = 2.190, *p* = 0.031), but not in cardiopulmonary fitness (VO2max; t(1, 111) = 0.605 *p* = 0.547).

### 3.2. OS Biomarkers Associated with Poor Sleep in CAD Patients

Serum 8-ISO concentrations were lower in patients with PSQI < 5 compared to those with PSQI ≥ 5 (27.887 ± 14.536 versus 37.961 ± 26.238, 95% confidence interval: −17.9 to −2.2, *p* = 0.014) (Figure 1). Our primary analysis revealed that baseline 8-ISO showed a difference between normal and poor sleepers in unadjusted (F(1111) = 6.212, *p* = 0.014, η_p_^2^ = 0.053) and adjusted (F(1108) = 4.684, *p* = 0.033, η_p_^2^ = 0.042) models controlling for demographics. In a model additionally adjusting for neuropsychiatric symptoms (anxiety and depressive symptoms based on HADS), no significant association was found (F(1106) = 1.240, *p* = 0.268, η_p_^2^ = 0.012). With other biomarkers of OS, no significant differences were found between the groups in unadjusted (F(1105) = 0.569, *p* = 0.453, η_p_^2^ = 0.005) and fully adjusted models in LPH levels (F(1100) = 0.296, *p* = 0.588, η_p_^2^ = 0.003) or unadjusted (F(1105) = 2.173, *p* = 0.143, η_p_^2^ = 0.020) and fully adjusted models of 8-ISO/LPH ratios (F(1100) = 1.421, *p* = 0.236, η_p_^2^ = 0.014).

### 3.3. OS Biomarkers Associated with Global Sleep Quality before and after Exercise Intervention

Baseline 8-ISO concentrations were positively associated with global PSQI scores before exercise intervention in models without covariates (adjusted R^2^ = 0.062, β = 0.265, *p* = 0.005) and with demographic covariates (adjusted R^2^ = 0.093, β = 0.215, *p* = 0.022), but not in the fully adjusted model (with demographic and neuropsychiatric symptom variables; adjusted R^2^ = 0.234, β = 0.099, *p* = 0.263).

In longitudinal analyses, baseline 8-ISO was not a significant predictor of termination global PSQI scores in models without covariates (adjusted R^2^ = 0.019, β = 0.176, *p* = 0.099), with demographic covariates (adjusted R^2^ = −0.002, β = 0.173, *p* = 0.114), or with demographic and neuropsychiatric symptom covariates (adjusted R^2^ = −0.016, β = 0.174, *p* = 0.128). All longitudinal models controlled for baseline PSQI scores.

### 3.4. OS Biomarkers Associated with Sleep Factors before and after Exercise Intervention

Among the OS biomarkers, 8-ISO was positively associated with Perceived Sleep Quality (Factor 2) and Daily Disturbances (Factor 3) at baseline in unadjusted and models adjusted for demographics (Table 2). In the fully adjusted model (with demographics and neuropsychiatric symptoms), 8-ISO was only associated with Daily Disturbances. The 8-ISO/LPH ratio was positively associated with Perceived Sleep Quality (Factor 2) in the unadjusted and fully adjusted models, but not with other sleep factors (Table 2). There were no significant associations between lipid peroxidation markers and termination sleep factors in the unadjusted models and models adjusted for demographics alone or for demographics and neuropsychiatric symptoms (Appendix A).

## 4. Discussion

The underlying pathway of sleep disturbances observed in patients with CAD is not well understood. OS has been proposed to underlie sleep disturbances in those patients. The present study found an association between lipid peroxidation and self-reported sleep quality in CAD patients. In patients who reported poor sleep quality (PSQI ≥ 5), concentrations of the late-stage lipid peroxidation marker 8-ISO were elevated compared to those who reported normal sleep. There were no differences in the early (LPH) or ratio of late- to early-stage OS biomarkers (8-ISO/LPH) between the two groups. Among the factors of sleep quality, 8-ISO was positively associated with Perceived Sleep Quality and Daily Disturbances while the 8-ISO/LPH ratio was associated with Perceived Sleep Quality. Neither baseline 8-ISO nor baseline 8-ISO/LPH were significant predictors of termination global PSQI scores and termination sleep factors.

### 4.1. Increased OS in CAD Patients with Worse Sleep Quality

Our analyses revealed greater baseline 8-ISO concentrations in those with worse sleep, which is in line with previous studies suggesting that sleep disturbances may be linked to increased OS [36,37]. Sleep has been proposed as a mechanism by which free radicals generated during wakefulness are eliminated [36] as levels of OS fluctuate along with the circadian rhythm [38]. Antioxidant enzymes, such as glutathione peroxidase and reductase, peak in the morning, whereas lipid peroxidation and melatonin peaks in the evening. As a result, OS accumulates during the day time and is eliminated at night [38]. Sleep disturbances disrupt the circadian rhythm and may therefore lead to an increase in free radicals and in OS [36,39]. In pre-clinical studies, animals deprived of rapid eye movement sleep demonstrated increased levels of ROS compared to controls; those levels were restored following five days of sleep recovery [40]. In night-shift workers, melatonin disruption was associated with an increased risk of OS, suggesting that sleep disturbances may be linked to a greater OS [41]. Our study did not find the early stage marker LPH to be significantly associated with sleep factor scores; this may be partly explained by the fact that the late-stage OS marker 8-ISO is more stable and can be more reliably measured [42]. There was also a lack of association between the ratio of late-stage marker (8-ISO) to the early-stage marker (LPH) and sleep quality. The ratios of late-stage marker (8-ISO) to the early-stage marker (LPH) may measure the balance of the redox system and reflect whether early stage markers are being converted to late-stage markers. As such, this finding may reflect a specific stage of oxidative stress, which has been previously reported in patients with bipolar disorder [43]. Consistent with that result, 8-ISO but not LPH was also found to be increased in patients with CAD [44] and bipolar disorder [45].

### 4.2. Determinants of Poor Sleep

Short sleep duration and poor sleep quality are suggested to lead to adverse cardiovascular events by affecting intermediate biological cardiovascular risk factors. Epidemiological studies revealed that poor sleep was linked to greater obesity, hypertension, higher blood pressure, and dyslipidemia [46]. In healthy individuals, short sleep duration increased the activation of the sympathetic nervous system and increased blood pressure [46]. In patients with sleep disorders, OS may be involved in promoting cardiovascular morbidity [47], possibly by mechanisms such as enhanced coagulability [48]. OS, specifically 8-ISO, is a major contributor to the pathophysiology of cardiovascular disease [42,49], and represents a potentially modifiable risk factor via antioxidant therapy or exercise; thus, our findings suggest a potential therapeutic mechanism in this at-risk population. Obstructive sleep apnea is highly prevalent in CAD populations and is associated with an increased risk of cardiovascular disorders [50,51]; a greater severity of obstructive sleep apnea is correlated with reduced antioxidant and greater systemic OS [52]. Furthermore, research has also suggested a bidirectional relationship between chronic obstructive pulmonary disease and poor sleep quality [53,54], both of which are associated with CAD [55]. Smoking status is a contributor to poor sleep in patients with chronic obstructive pulmonary disease [53] and is a well-established risk factor of CVD and mortality [56]. Smoking is proposed to affect sleep quality by eliciting abnormal sleep architecture, such as sleep deprivation and disruption [57,58]. Smoking may alter the release of neurotransmitters that modulate the sleep–wake cycle if nicotine levels drop during sleep [59] In addition, the inflammatory marker C-reactive protein (CRP) has been of interest as it can be measured in many laboratories and has a close association to cardiovascular disease [60,61]. There is currently mixed literature on whether CRP may be increased [62,63] or decreased [61] in obstructive sleep apnea patients and whether treatment with continuous positive airway pressure is associated with improved CRP levels [60,62,64]. Our study did not record history or presence of chronic obstructive pulmonary disease; however, participants in this study are long-term former smokers (average 17 years), which could modulate the development of chronic obstructive pulmonary disease and poor sleep [65].

### 4.3. Factors Contributing to Lipid Peroxidation Development

Although not measured in this study, various external factors in addition to CAD may impact the concentration of early and late lipid peroxidation markers. CAD and cancer are independent risk factors for each other [66]; certain anticancer drugs have been established to induce lipid peroxidation [67]. For example, vinca alkaloids deplete intracellular GSH levels while increasing intracellular ROS production [68], and epirubicin and doxorubicin contain an anthracycline skeleton that stimulate ROS production leading to greater OS [67,69,70]. Lastly, lifestyle choices such as frequent alcohol abuse lowers antioxidant capacity. During the metabolism of alcohol to aldehyde, free radicals are produced and can react directly with hepatic proteins via alkylation to produce lipid peroxidation [71,72,73].

The results are consistent with the relationship between cardiometabolic health and oxidative stress levels in healthy individuals. Results from a study conducted by Kanagasabai et al., showed that, in normal (or fair quality) sleepers, OS levels were at an optimal level compared to poor and very poor quality sleepers, suggesting a relationship between OS and poor sleep [6,74]. Furthermore, there is a positive relationship between body mass index and lipid peroxidation concentrations in heathy populations [75]. In healthy individuals, free radicals play a role in biological processes such as intracellular elimination of bacteria via phagocytes and cellular signalling (redox signalling) [76,77]. At regulated and low amounts, ROS have beneficial effects in maintaining homeostasis and cellular functioning [77].

### 4.4. Exercise and OS

Intensity, mode, and duration of exercise are key modulators of metabolic, cardiovascular, and central nervous systems [78,79,80]. Regular moderate-intensity exercise enhances endogenous antioxidant levels [81] and reduces lipid peroxidation in animals [82], whereas in schizophrenia patients, the duration of moderate physical activity per week was positively correlated with sleep quality [83]. In particular, acute moderate-intensity exercise improves sleep more than high-intensity aerobic exercise or high-intensity resistance exercise [84]. Although the time at which exercise is performed does not affect sleep parameters in healthy individuals [85,86], the mode of exercise is thought to be of importance as aerobic (80% at maximal aerobic capacity) and resistance training (80% of maximal load) improve sleep quality [87]. This is aligned with the current study protocol that used a moderate-intensity aerobic and resistance training program to achieve a maximum of 80% VO2max. In our findings, baseline 8-ISO levels were not associated with termination global PSQI scores, which may be explained by the more gradual change in OS seen in chronic exercise interventions compared to acute exercise [88]. This theory is consistent with meta-analyses showing that, although acute exercise has a beneficial effect on factors of sleep such as sleep disturbances, those effects are small and inconsistent [89,90]. Furthermore, neuropsychiatric symptoms may mediate the relationship between exercise and sleep as healthy individuals who exercised reported lower symptoms of depression and anxiety, and better sleep quality [91].

### 4.5. Strengths

This study used validated questionnaires and biomarkers to investigate the relationship between sleep disturbances and OS in a well-characterized cardiovascular disease population. While several studies investigated sleep disorders and cardiovascular mortality, fewer studies have investigated sleep symptoms in CAD patients [33]. Sleep symptoms, as opposed to sleep disorders, are of clinical interest as they indicate sub-clinical disturbances [92] and may provide a specific symptom target for treatment. In this cohort of CAD patients, over 50% of patients reported experiencing poor sleep quality, and our results show that OS was increased at this subclinical symptomatic stage. Our study also found sleep factors may be preferred over the global PSQI score as it provided more information on the nature and type of sleep problems, which is important for the selection of treatment [24].

### 4.6. Limitations

There are a few limitations to this study. First, our study design did not allow us to conclude any direct relationships between OS and sleep disturbances. Second, this study did not account for concomitant medications that may affect both OS and sleep, such as stimulants and decongestants. Third, although the effects of exercise timing on sleep are ambiguous [84,89] exercise timing may potentially disrupt sleep, and future studies could investigate whether exercise timing affects OS. Fifth, this study only included moderate-intensity exercise training, whereas previous literature has found that high-intensity training promoted better sleep Appendix A [84]. Lastly, our study did not analyze changes in anxiety and depression scores following exercise treatment, which may affect the association between OS and termination sleep scores. It is important to mention that sleep quality in CAD populations may be related to various other biomarkers that were not included in this study.

## 5. Conclusions

This study found associations between self-reported sleep quality and OS in older CAD patients. This relationship is consistent with the suggestion that sleep disturbances in CAD contribute to increased OS. Importantly, late-stage lipid peroxidation markers were associated with Perceived Sleep Quality and Daily Disturbances. Future studies should investigate the possible benefits of antioxidant therapy in combination with physical exercise for patients reporting poor sleep quality.

## Figures and Tables

**Figure 1 brainsci-12-01070-f001:**
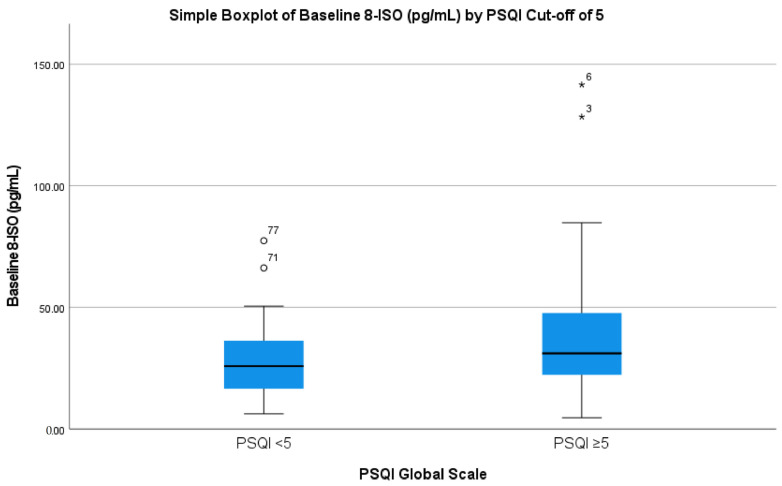
Box plot displaying the median and interquartile range of unadjusted 8-ISO levels in CAD patients with normal and poor sleep (score of at least of 5 on the PSQI scale). Symbols (* and ^o^) indicate extreme outliers (>3SD). Error bars represent standard error of the mean.

**Table 1 brainsci-12-01070-t001:** Participant characteristics in total population and those with normal sleep compared with poor sleep quality.

Characteristic	All (*N* = 113) Mean (SD) or *n* (%)	Normal Sleep Quality (*n* = 54) Mean (SD) or *n* (%)	Poor Sleep Quality (*n* = 59) Mean (SD) or *n* (%)	Test-Statistic **	*p*-Value
Socio-demographic					
Age	63.7 (6.4)	64.8 (6.6)	62.6 (6.1)	t = 1.9	0.06
Sex (Male)	96 (85.0%)	48 (42.5%)	48 (42.5%)	χ^2^ = 1.3	0.3
Married/Common-law	89 (78.8%)	41 (36.3%)	48 (42.5%)	χ^2^ = 0.5	0.5
Total Years Smoked (*n* = 110, 53/57)	14.8 (16.4)	14.3 (17.6)	15.3 (15.3)	t = −0.3	0.7
Currently Smoking	3 (2.7%)	2 (1.8%)	1 (0.9%)	- ^a^	0.6
Number of caffeinated beverages per day (*n* = 112, 53/59))	2.2 (2.1)	2.5 (2.4)	2.0 (1.7)	t = −1.3	0.2
Cardiorespiratory					
Maximum Systolic (*n* = 112, 59/53)	171.5 (23.2)	172.1 (23.5)	170.9 (23.1)	t = 0.3	0.8
Maximum Diastolic (*n* = 112, 59/53)	78.5 (11.1)	78.4 (10.1)	78.7 (12.0)	t = −0.2	0.9
Body mass index (kg/m^2^)	29.2 (5.2)	28.9 (4.9)	29.5 (5.6)	t = −0.7	0.5
Hypercholesterolemia	31 (27.4%)	14 (12.4%)	17 (15.0%)	χ^2^ = 0.1	0.7
Hypertension	106 (93.8%)	51 (45.1%)	55 (48.7%)	- ^a^	1
Diabetes	20 (17.7%)	10 (8.8%)	10 (8.8%)	χ^2^ = 0.05	0.8
Sleep Apnea (*n* = 112, 58/54)	21 (18.6%)	7 (6.3%)	14 (12.5%)	χ^2^ = 2.3	0.1
Concomitant Medications					
Beta-Blocker	89 (78.8%)	47 (41.6%)	42 (37.2%)	χ^2^ = 4.2	0.04 *
ACE-Inhibitor	62 (54.9%)	29 (25.7%)	33 (29.2%)	χ^2^ = 0.06	0.8
Statin	113 (100%)	54 (100%)	59 (100%)	-	-
Neuropsychiatric symptoms and OS markers					
HADS (Anxiety)	4.0 (3.2)	2.8 (2.3)	5.1 (3.5)	t = −4.1	<0.001 *
HADS (Depression)	2.5 (2.5)	1.6 (1.8)	3.4 (2.8)	t = −3.9	<0.001 *
Baseline 8-ISO (pg/mL)	33.1 (22.0)	27.9 (14.5)	38.0 (26.2)	t = −2.6 ^b^	0.01 *
Baseline LPH (uM) (*n* = 107, 53/54)	41.7 (31.4)	44.1 (30.9)	39.5 (32.0)	t = 0.8	0.5
Baseline 8-ISO/LPH (*n* = 107, 53/54)	2.8 (7.2)	1.8 (3.8)	3.8 (9.3)	t = −1.5 ^b^	0.1
Sleep Quality Scores					
Baseline PSQI	5.8 (4.0)	2.5 (1.0)	8.9 (3.2)	t = −14.3	<0.001 *
Termination PSQI (*n* = 95, 49/46)	4.6 (3.3)	4.4 (3.2)	4.8 (3.5)	t = −0.6	0.5

* Significant at *p* ≤ 0.05; ** Significance at *p* ≤ 0.01; ^a^ Fischer’s exact test used; ^b^ equal variances not assumed; Percentages refer to percent of total participants with value. Body Mass Index (BMI), Hospital Anxiety and Depression Scale (HADS), Angiotensin-converting enzyme (ACE) inhibitors.

**Table 2 brainsci-12-01070-t002:** Results of linear regression models using 8-ISO and 8-ISO/LPH as predictor variables for baseline sleep factors.

	Overall Model					Association with Lipid Peroxidation Markers			
	Adjusted R^2^	Mean Square	F	*p*-Value	B	SE	β	t	*p*-Value
Model 1—unadjusted									
8−ISO (*n* = 113)									
Sleep Efficiency	0.007	3.986	1.774	0.186	0.009	0.006	0.125	1.332	0.186
Perceived Sleep Quality	0.061	13.983	8.314	0.005 *	0.016	0.006	0.263	2.883	0.005 *
Daily Disturbances	0.142	6.502	19.593	<0.001 *	0.011	0.002	0.387	4.426	<0.001 *
8−ISO/LPH (*n* = 107)									
Sleep Efficiency	−0.003	1.478	0.701	0.404	0.016	0.020	0.081	0.837	0.404
Perceived Sleep Quality	0.033	7.688	4.614	0.034 *	0.038	0.017	0.205	2.148	0.034 *
Daily Disturbances	−0.008	0.078	0.207	0.650	0.004	0.008	0.045	0.455	0.650
Model 2—adjusted for demographics **									
8-ISO (*n* = 113)									
Sleep Efficiency	0.011	2.969	1.327	0.264	0.006	0.007	0.091	0.947	0.346
Perceived Sleep Quality	0.069	5.162	3.097	0.019 *	0.014	0.006	0.222	2.388	0.019 *
Daily Disturbances	0.228	2.770	9.275	<0.001 *	0.009	0.002	0.329	3.875	<0.001 *
8-ISO/LPH (*n* = 107)									
Sleep Efficiency	−0.015	1.284	0.602	0.662	0.014	0.021	0.070	0.687	0.493
Perceived Sleep Quality	0.031	3.070	1.838	0.127	0.029	0.018	0.161	1.616	0.109
Daily Disturbances	0.111	1.430	4.284	0.003 *	−0.002	0.008	−0.018	−0.189	0.851
Model 3—adjusted for demographics and neuropsychiatric symptoms ***									
8-ISO (*n* = 113)									
Sleep Efficiency	0.050	4.293	1.998	0.072	0.002	0.007	0.029	0.294	0.770
Perceived Sleep Quality	0.221	7.444	5.051	<0.001 *	0.007	0.006	0.120	1.314	0.192
Daily Disturbances	0.445	3.428	15.957	<0.001 *	0.005	0.002	0.190	2.530	0.013 *
8-ISO/LPH (*n* = 107)									
Sleep Efficiency	0.120	6.289	3.399	0.004 *	0.010	0.020	0.051	0.531	0.596
Perceived Sleep Quality	0.246	8.789	6.764	<0.001 *	0.033	0.016	0.181	2.024	0.046 *
Daily Disturbances	0.410	2.915	13.138	<0.001 *	−0.003	0.007	−0.039	−0.495	0.622

* Significant at *p* ≤ 0.05. ** Demographics include age, sex, and BMI; *** Neuropsychiatric symptoms as measured by HADS (Hospital Anxiety and Depression Scale).

## Data Availability

Data are not publicly available due to patient confidentiality but may be presented upon reasonable request.

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
