# Peer review of "The Relationship between Oxidative Stress and Subjective Sleep Quality in People with Coronary Artery Disease"

_brainsci, 2022, doi:10.3390/brainsci12081070_

Round 1

Reviewer 1 Report

This paper entitled “The relationship between oxidative stress and subjective sleep quality in people with coronary artery disease” describes a case study to evaluate the potential relationship between oxidative stress (OS) and sleep, as sleep disorders often happen in coronary artery disease (CAD). The authors measured concentration of lipid hydroperxides (LPH) and 8-isoprostane (8-ISO) in blood samples from CAD patients and used self-reported PSQI to assess sleep quality. The authors found a corelated relationship between oxidative stress and sleep efficiency and daily disturbances. Moreover, the author found that 8-ISO may serve as a potential biomarker for sleep quality.

It has been shown that patients with cardiovascular diseases often exhibit sleep disorders, but the relationship between oxidative stress and sleep remains unknown, which this paper try to address. Overall, this paper confirmed a relationship between OS and sleep, and identification of 8-ISO can be used as a biomarker for sleep is meaningful and important for both clinical and biomedical study. However, the following points need to be addressed prior to publication:

1.     Whether cardiac rehabilitation has positive effect on sleep is not clear. Line 141 claimed a significant improvement of sleep after 24-week exercise, but data is missing. Also, is the improvement of sleep related to oxidative stress status? Were the amounts of 8-ISO or LPH compared before/ after exercise?

2.     Do 8-ISO and LPH express differently at the daytime (wake) from nighttime (sleep)? It’s not clear when the blood samples were collected.

3.     No additional experiment is required, but more introduction/discussion needed for LPH and 8-ISO in normal people and potential function in sleep.  

Minor comments:

1.     There are some formatting issues in the paper. For example, indentation of the result section is different from the rest.

2.     Line 149: suggest to delete it.

3.     Figure 1: explain what do B1 and B2 represent, and what does  * indicate? What’s error bar stand for?

Author Response

Please see the attachment under reviewer 1 

Reviewer 2 Report

This is an interesting paper investigating the association between oxidative stress, sleep quality and coronary artery disease in people under exercise.

The paper is well-written and of interest for the readers. However, several minor changes should be made before considering it for publication.

In the abstract section, I recommend to rephrase the sentence aboout "sleep disorders  (...) predict ... poor prognosis". Sleep disorders can predict prognosis but not "poor". 

Which other factors are potentially influencing concentrations of early and late lipid peroxidation products? 

In the introduction section, the authors report that Sleep-wake disturbances are associated with cardiovascular disease. Which is the direction of the relationship? Is bidirectional?

Is oxidative stress the only variable capable of explaining cardiovascular risk? The authors should emphasize that their hypothesis is a proxy. 

Lines 42-43: These should be placed before aims, at the end of the introduction section.

The study design should be clarified in the methods section as well as in the abstract section. 

At the beginning of the Results section, the authors report that Data from 120 study participants were obtained, and then they  give the NCT number. It should be placed in the methods section.

In the discussion section, the authors should not repeat the results. In the first paragraph they are repeating that late-stage lipid peroxidation levels were higher in patients with higher than 5 PSQI scores. It should be rephrased. 

Which other factors are influencing the association between sleep disorders and CV risk? Is there any other biomarkers? It should be added in the discussion when analyzing results.

I recommend to expand the discussion and number of citations about the association between sleep and moderate or high-intensity training. This is crucial to better understand why the authors selected a type of intervention. 

A strengths and limitation section is necessary. I would add a discussion about the limitations of the administered scales (quality, subjective) and the biomarkers used to assess oxidative stress.

Author Response

Please see attachment under reviewer 2 

Round 2

Reviewer 1 Report

I'm happy to see that most of my questions have been addressed by the authors, and I don't have further questions or comments prior to its publication.